# Physical exercise and chronic pain in university students

**Michael Grasdalsmoen**[1], **Bo Engdahl**[2], **Mats K. Fjeld**[2], **Ólöf A. Steingrímsdóttir**[2], **Christopher S. Nielsen**[2,3], **Hege R. Eriksen**[1], **Kari Jussie Lønning**[4,5], **Børge Sivertsen** [6,7,8] *

**1** Department of Sport, Food and Natural Sciences, Western Norway University of Applied Sciences, Bergen, Norway, **2** Department of Chronic Diseases and Ageing, Norwegian Institute of Public Health, Oslo, Norway, **3** Department of Pain Management and Research, Oslo University Hospital, Oslo, Norway, **4** The Norwegian Medical Association, Oslo, Norway, **5** The Student Welfare Association of Oslo and Akershus (SiO), Oslo, Norway, **6** Department of Health Promotion, Norwegian Institute of Public Health, Bergen, Norway, **7** Department of Research & Innovation, Helse Fonna HF, Haugesund, Norway, **8** Department of Mental Health, Norwegian University of Science and Technology, Trondheim, Norway

* borge.sivertsen@fhi.no

**Data Availability Statement:** The datasets for this article are not publicly available because of privacy regulations from the Norwegian Regional Committees for Medical and Health Research Ethics (REC). Requests to access the datasets

## Abstract

### Background

Physical inactivity and chronic pain are both major public health concerns worldwide. Although the health benefits of regular physical exercise are well-documented, few large epidemiological studies have investigated the association between specific domains of physical exercise and chronic pain in young adults. We sought to investigate the association between frequency, intensity and duration of physical exercise, and chronic pain.

### Methods

Data stem from the SHoT2018-study, a national health survey for higher education in Norway, in which 36,625 fulltime students aged 18–35 years completed all relevant questionnaires. Chronic pain, defined according to the International Classification of Diseases 11th Revision (ICD-11), was assessed with a newly developed hierarchical digital instrument for reporting both distribution and characteristics of pain in predefined body regions. Physical exercise was assessed using three sets of questions, measuring the number of times exercising each week, and the average intensity and the number of hours each time.

### Results

The majority (54%) of the students reported chronic pain in at least one location, and the prevalence was especially high among women. The overall pattern was an inverse dose-response association between exercise and chronic pain: the more frequent, harder or longer the physical exercise, the lower the risk of chronic pain. Similar findings were generally also observed for the number of pain locations: frequent exercise was associated with *fewer* pain locations. Adjusting for demographical, lifestyle factors and depression had little effect on the magnitude of the associations.

should be directed to the Norwegian Institute of Public Health (Datatilgang@fhi.no). Guidelines for access to SHoT data are found at https://www.fhi.no/en/more-access-to-data. Approval from REC (https://helseforskning.etikkom.no) is a pre-requirement.

**Funding:** SHoT 2018 has received funding from the Norwegian Ministry of Education and Research (2017) and the Norwegian Ministry of Health and Care Services (2016) to KJL. The funders had no role in study design, data collection and analysis, decision to publish, or preparation of the manuscript.

**Competing interests:** The authors have declared that no competing interests exist.

## Conclusion

Given the many health benefits of regular exercise, there is much to be gained in facilitating college and university students to be more physically active, ideally, thru a joint responsibility between political and educational institutions. Due to the cross-sectional nature of the study, one should be careful to draw a firm conclusion about the direction of causality.

## Introduction

Chronic pain is a major public health concern worldwide, with significant impact on both an individual and a socioeconomic level. While prevalence estimates vary, reviews suggest that chronic pain affects about one-third of the general population [1, 2], and pain was recently highlighted as one of the leading causes of disability by the Global Burden of Disease (GBD) study [3].

Individual variations, including expectations and sensitization [4] caused by a complex interplay between genetic and environmental factor, influence the experience of pain [5]. There is some evidence suggesting a link between a physical inactivity and the development of chronic pain [6, 7]. Several population-based studies have indicated that physical exercise may indeed reduce the risk of chronic pain [8–10]. However, the literature in this field remains mixed, with conflicting evidence regarding a possible association between leisure-time physical activity and pain in the general population [11, 12]. Moreover, most studies in this field are in samples of middle-aged and older adults, with less knowledge about this possible link in younger adults. On the one hand, students pursuing higher education are generally in good health, and many universities try to facilitate their students to be physically active. On the other hand, college and university students may also be prone to inactivity, as many spend up to 10 hours a day in environments characterized by prolonged sitting [13].

Recent evidence shows that the world´s total physical activity level is on the decline across all age groups [14]. Alongside the large public health burden of chronic pain, and what may be labelled the global obesity epidemic [15], it is particularly important to examine the link between physical exercise and chronic pain in younger samples. This group has the potential to live a healthy life and stay in the workforce for many years ahead.

A challenge with the existing research in this field is the heterogeneity in assessment methods, regarding both operationalizations of physical exercise, as well as chronic pain. This diversity in methodology may lead to inaccurate estimates of both prevalence rates and magnitude of associations [2]. In the current study, we aim to improve on these limitations by employing both a well-validated measure on physical exercise, as well as including a newly developed hierarchical digital instrument for reporting both distribution and characteristics of pain in predefined body regions (the Graphical Index of Pain (GRIP) [16].

In terms of potential mechanisms that may account for the association between physical activity and chronic pain, both demographic factors, including gender [17, 18], ethnicity [19], and socioeconomic status [20], as well as health behaviors (alcohol use [21, 22]), sleep problems [23, 24], and depression [25] have previously shown associations to both exercise and pain in young adulthood. These factors are, therefore, essential to account for when examining the link between exercise and chronic pain.

Based on these considerations, using data from a large national study from 2018 of all Norwegian college and university students, we investigated the level of chronic pain as among male and female students in higher education. We also examined the association between the

frequency, intensity and duration of physical exercise, and chronic pain, and if sociodemographic, lifestyle or other health factors could explain any of the observed associations.

## Materials and methods

### Procedure

The current paper used data from the SHoT2018 study (*Students' Health and Wellbeing Study)*, a large national survey of students enrolled in higher education in Norway, initiated by the three largest student welfare organizations. The SHoT2018 is a comprehensive survey of several domains of health, quality of life and academic functioning, and was collected electronically through a web-based platform. Details of the study has been published elsewhere [26], but in short, SHoT2018 was conducted between February 6 and April 5, 2018, and invited all fulltime Norwegian students pursuing higher education, both in Norway and abroad. In all, 162,512 students fulfilled the inclusion criteria, of whom 50,054 students completed the online questionnaires, yielding a response rate of 30.8%. After first completing the main SHoT2018 questionnaire, all participating students were redirected to the GRIP questionnaire (described below), which 36,625 (73.2%) completed, yielding an overall 22.5% response rate. During the GRIP data collection, the data server had some downtime due to heavy traffic. This may explain why fewer participants completed the GRIP questionnaire. Validated Norwegian translation (i.e. physical exercise, sleep duration and AUDIT) were used, and questionnaires not previously translated into Norwegian were translated, and then back-translated to the original language (English), to ensure accuracy.

### Independent variables

**Physical exercise.**   First, the following brief definition of physical exercise was presented to the students: "*With exercise, we mean that you, for example, go for a walk, go skiing, swim or take part in a sport*". Exercise was then assessed using three sets of questions, including the average number of times exercising each week, and the average intensity and average hours each time [27]: 1*) "How frequently do you exercise*?*"* (Never, Less than once a week, Once a week, 2–3 times per week, Almost every day); 2) *"If you exercise as frequently as once or more times a week*: *How hard do you push yourself*? (I take it easy without breaking into a sweat or losing my breath, I push myself so hard that I lose my breath and break into a sweat, I push myself to near-exhaustion); and 3) *"How long does each session last*?*"* (Less than 15 minutes, 15–29 minutes, 30 minutes to 1 hour, More than 1 hour". This 3-item questionnaire has previously been used in the large population-based Nord-Trøndelag Health Study (HUNT) [27, 28]. In the current study, the response options "Never" and "Less than once a week" were combined for the frequency item, constituting the reference category. For the duration item, the response options "Less than 15 minutes" and "15–29 minutes" were also combined for the same reasons. Detailed information on the physical exercise items in the SHoT2018 study has been published elsewhere [29]. Previous validation studies [27, 28] have demonstrated moderate correlations between these questionnaire items and direct measurement of $VO_2max$ during maximal work on a treadmill (r = 0.43[frequency], r = 0.40 [intensity] and r = 0.31 [duration]), with ActiReg [30, 31] (an instrument that measures PA and energy expenditure), and with the International Physical Activity Questionnaire [32].

### Dependent variables

**The Graphical Index of Pain (GRIP).**   GRIP is a hierarchical digital body map designed to assess pain and pain-related characteristics [16]. The instrument consists of 10 first-tier

regions (head, neck, left arm, right arm, upper and lower back, left leg, right leg, chest, abdomen, genitals/pelvic floor/urethra/anus) followed by anatomical sites at second-tier (167 loci among men and 168 loci among women). Participants were asked to report pain experienced within the last 4 weeks, omitting brief transient pain. Pain characteristics were reported for each of the marked first-tier regions, i.e. pain duration, episode frequency, episode duration, intensity, how bothersome the pain was, and interference with daily activities and sleep. Women were instructed not to report menstrual pain. Instructions and questions in GRIP were put in Norwegian. Translation to English was made by a certified translator, but back translation is still in process [16].

**Chronic pain.** The definition of chronic pain was based upon the ICD-11 criteria, with pain persisting or recurring for longer than 3 months [33]. In GRIP, subjects reported the time since first onset of pain. The options were: "Less than 4 weeks", "1–2 months", "3–5 months", "6–11 months", "1–2 years", "3–5 years", "More than 5 years (asked about the age of onset)". Hence, the chronic pain definition in the present study was pain experienced within the last 4 weeks in at least one of ten first tier loci with ≥3 months duration. For purposes of the present study, the GRIP was used to produce several heat maps to visualize the prevalence and distribution of chronic pain.

**Moderate to severe chronic pain.** The ICD-11 definition of moderate to severe chronic pain is based on three pain-related parameters [33]: a) pain intensity, b) pain-related distress and c) task interference. The assessment may be graded on a 100-mm visual analogue scale (VAS) (4). The participants in SHoT2018 were asked to grade the following pain characteristics on a VAS (from 0 to 10): a) pain intensity (anchors: No pain/The strongest imaginable pain), b) bothering as a proxy of pain-related distress (anchors: No bother/The greatest imaginable bother), and c) impact on activity in daily activities, as a proxy of task interference (anchors: Not at all /Can't do anything) (4). We regarded that moderate to severe chronic pain was present in subjects reporting pain within the last 4 weeks in at least one of 10 first tier body regions, with the onset of ≥3 months, and with pain intensity of VAS ≥4, bothering of VAS ≥4, and impact on daily activities of VAS ≥4.

## Control variables

**Sociodemographic information.** All participants reported their gender, age and relationship status (coded as single versus married/partner or girl-/boyfriend). Economic activity was coded dichotomously according to self-reported annual income (before tax and deductions, and not including loans and scholarships): "economically active" (annual income > 10,000 NOK) versus "economically inactive" (≤ 10,000 NOK). Finally, participants were categorized as an immigrant if either the participants or one or both of his/her parents were born outside Norway.

**Body Mass Index (BMI).** BMI was calculated based on self-reported body weight (kg) divided by self-reported squared height ($m^2$), and categorized as underweight (BMI < 18.5), normal weight (BMI 18.5–24.9), overweight (BMI 25.0–29.9) and obesity (BMI ≥ 30). Trend data on overweight and obesity from the SHoT studies have been published elsewhere [29].

**Sleep duration.** The participants' self-reported usual bedtime and wake up time were indicated in hours and minutes, and data were reported separately for weekdays and weekends. Time in bed (TIB) was calculated as the difference between bedtime and wake up time. Sleep onset latency (SOL) and wake after sleep onset (WASO) were also indicated separately for weekdays and weekends in hours and minutes. Sleep duration was defined as TIB minus SOL and WASO. More detailed information about the sleep inventory in SHoT2018 has been published elsewhere [34].

**Alcohol-related problems.**  Alcohol-related problems were assessed by the Alcohol Use Disorders Identification Test (AUDIT), which is a widely used instrument developed by the World Health Organization to identify risky or harmful alcohol use [35, 36]. The 10-item AUDIT includes items for measuring the frequency, typical amount and episodic heavy drinking frequency (items 1–3), alcohol dependence (items 4–6), and problems related to alcohol consumption (items 7–10) [37]. The AUDIT score ranges from 0 to 40. More information about the AUDIT in the SHoT surveys has been published elsewhere [38].

**Depression.**  Self-reported depressive disorder was assessed from a pre-defined list of several common somatic and mental conditions/disorders adapted to fit this age-cohort. The list was based on a similar operationalization used in previous large population-based studies (the HUNT study [39]) and included several subcategories for most conditions/disorders (not listed here). For mental disorders, the list comprised the following specific disorders/ group of disorders: ADHD, anxiety disorder, autism/Asperger, bipolar disorder, depression, PTSD (posttraumatic stress disorder), schizophrenia, personality disorder, eating disorder, Tourette's syndrome, obsessive-compulsive disorder (OCD), and other. The list contained no definition of the included disorders/conditions. In the current study, only depressive disorder was included.

## Statistical analyses

The heat maps were created in R (version 3.6.1; https://www.r-project.org) with functions to create vector graphics to color loci of the GRIP images, based on values in the input data matrix. IBM SPSS Statistics 25 for Windows (SPSS Inc., Chicago, IL) was used for the other analyses. Pearson's chi-squared tests were used to examine differences in the prevalence of pain by physical exercise level, stratified by gender. Logistic regression models were computed to obtain effect-size estimates for the dichotomous dependent variables. Results are presented as odds-ratios (ORs) with 95% confidence intervals. We computed one unadjusted and two adjusted models. In the first block we controlled for socio-demographic factors (categorical), body-mass index (continuous), alcohol use and problems (AUDIT continuous sum score), and sleep duration (continuous). In the second block (fully adjusted model) we additionally adjusted for self-reported depression. Estimated marginal means (EMM) were also computed to examine exercise frequency against number of pain loci, adjusting for age. Missing values were handled using listwise deletion.

## Ethics

All procedures involving human subjects/patients were approved by the Regional Committee for Medical and Health Research Ethics in Western Norway (no. 2017/1176 [SHOT2018]). Electronic informed consent was obtained after the participants had received a detailed introduction to the study.

## Results

### Sample characteristics

In all, 36625 students (67.2% women [n = 24600] and 32.8% men [n = 12025]) with a mean age of 23.2 years, completed both the main SHOT2018 questionnaire and the additional GRIP instrument. Approximately half of the students were single, 87% had no additional income besides students' loan and scholarships, while 8% were of non-Norwegian ethnicity. More details of sociodemographic and clinical characteristics are listed in Table 1.

Table 1. Sociodemographic and clinical characteristics of the study sample.

| | Women | | Men | | | Total sample | |
|---|---|---|---|---|---|---|---|
| | mean / n | SD / % | mean / n | SD / % | p-value$^§$ | mean / n | SD / % |
| Age, mean (SD) | 23.2 | (3.3) | 23.6 | (3.3) | < .001 | 23.2 | (3.3) |
| Marital status, n (%) | | | | | < .001 | | |
| Single | 13004 | (52.9%) | 5323 | (44.3%) | | 18327 | (50.1%) |
| Married/partner/girl- or boyfriend | 11570 | (47.1%) | 6683 | (55.7%) | | 18253 | (49.9%) |
| Immigrant status, n (%) | | | | | .292 | | |
| Ethnic Norwegian | 22644 | (92.0%) | 11056 | (91.9%) | | 33700 | (92.0%) |
| Immigrant | 1956 | (8.0%) | 969 | (8.1%) | | 2925 | (8.0%) |
| Economic activity, n (%) | | | | | < .001 | | |
| Active | 2836 | (12.0%) | 1874 | (15.9%) | | 4710 | (13.3%) |
| Inactive | 20843 | (88.0%) | 9894 | (84.1%) | | 30737 | (86.7%) |
| Body-mass index category, n (%) | | | | | | | |
| Underweight | 973 | (4.1%) | 424 | (2.1%) | | 1215 | (3.4%) |
| Normal weight | 15604 | (65.5%) | 7234 | (61.6%) | | 22838 | (64.2%) |
| Overweight | 5054 | (21.2%) | 3355 | (28.6%) | | 8409 | (23.7%) |
| Obese | 2189 | (9.2%) | 7904 | 7.7%) | | 3090 | (8.7%) |
| Sleep duration, mean (SD) | 7:26 | (1:24) | 7:24 | (1:24) | .047 | 7:25 | (1:24) |
| AUDIT sum score, mean (SD) | 6.7 | (4.4) | 8.1 | (5.1) | < .001 | 7.3 | (4.7) |
| Depression, n (%) | 2943 | (12.0%) | 912 | (7.6%) | < .001 | 3855 | (10.5%) |
| ICD-11 Chronic pain, # body regions, n (%) | | | | | < .001 | | |
| None | 9854 | (40.1%) | 6914 | (57.5%) | | 16768 | (45.8%) |
| 1 body region | 4900 | (19.9%) | 2355 | (19.6%) | | 7255 | (19.8%) |
| 2 body regions | 4118 | (16.7%) | 1473 | (12.2%) | | 5591 | (15.3%) |
| 3 or more body regions | 5728 | (23.3%) | 1283 | (10.7%) | | 7011 | (19.1%) |
| ICD-11 Chronic pain moderate to severe, # body regions, n (%) | | | | | < .001 | | |
| None | 19894 | (80.9%) | 11172 | (92.9%) | | 31066 | (84.8%) |
| 1 body region | 1846 | (7.5%) | 434 | (3.6%) | | 2280 | (6.2%) |
| 2 body regions | 1263 | (5.1%) | 235 | (2.0%) | | 1498 | (4.1%) |
| 3 or more body regions | 1597 | (6.5%) | 184 | (1.5%) | | 1781 | (4.9%) |

$^§$ p-values based on overall Chi-squared analyses (categorical variables) or independent samples t-test (continuous variables)

## Physical exercise and chronic pain

About half (54.2%) of students reported at least one chronic pain location. Nearly one in five students (19.1%) reported three or more chronic pain locations. There were large gender differences in the prevalence of pain, with female students reporting significantly more pain compared to the male students (Table 1). Fig 1 shows the prevalence and distribution of chronic pain stratified by the average weekly frequency of physical exercise in men and women. There was an inverse dose-response association between exercise frequency and chronic pain: the more frequent exercise, the less chronic pain. Table 2 provides more details of the association between the three exercise items (frequency, intensity, duration) and chronic pain stratified by gender. Compared to exercising never or less than once a week, female students who exercised almost every day were 23% less likely (OR = 0.77) to have chronic pain. Adjusting for potential demographical and clinical confounders only slightly attenuated this association (OR = 0.85, 95% CI: 0.77–0.83). Similarly, exercising 2–3 times per week and once a week decreased the odds of chronic pain compared to exercising less than once a week (Table 2). In terms of the

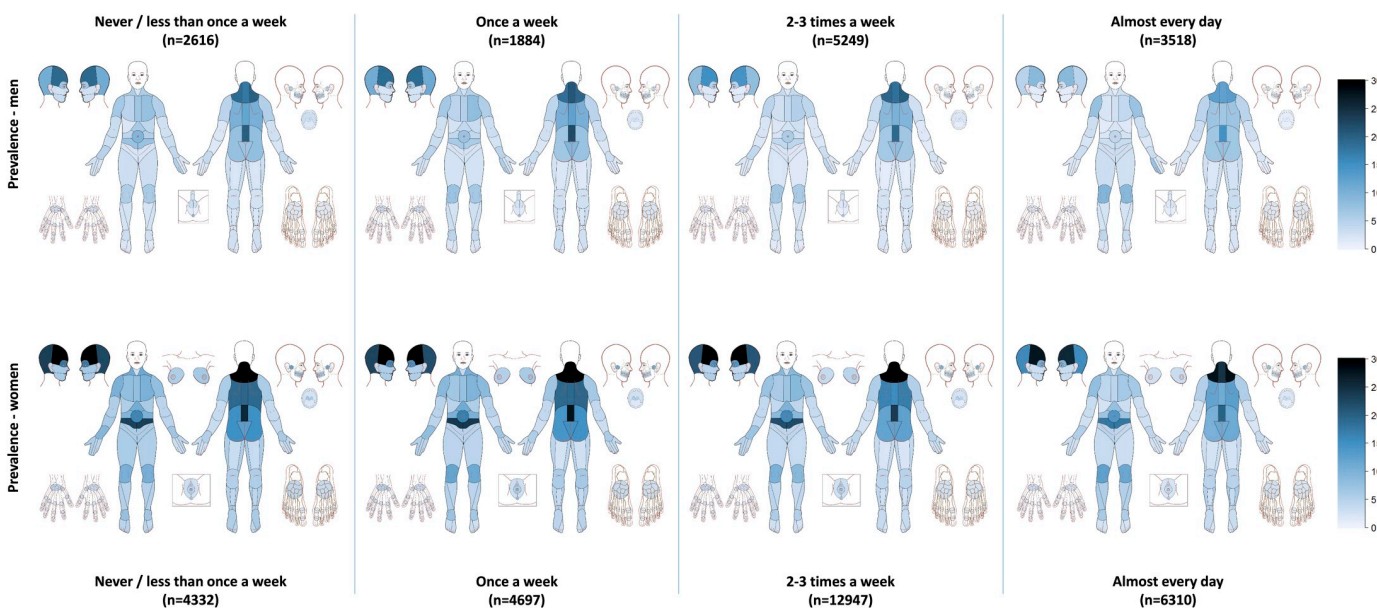

**Fig 1. Prevalence and distribution of chronic pain by the average weekly frequency of physical exercise in men and women.**

*intensity* of the physical exercise among females, there were no clear or strong trends regarding the odds of chronic pain. In contrast, the *duration* of the exercise was inversely associated with reporting more pain: female students exercising more than 1 hour a week were around 20% less likely to report chronic pain compared to students exercising hour less than 30 minutes a week (fully adj. OR = 0.83, 95% CI: 0.75–0.92).

Similar trends were observed among male students, but the beneficial effects were overall higher (lower ORs) than for females, especially regarding the frequency of physical exercise. In contrast to female students, there was a significant inverse dose-response relationship between exercise intensity and chronic pain among male students. This pattern was also observed for exercise duration; the longer duration of the exercise, the lower odds for chronic pain. Overall, adjusting for the potential confounding factors had little effect on the magnitude of the associations (Table 2).

## Physical exercise and moderate to severe chronic pain

When examining the association between the physical exercise items and odds of *moderate-to-severe* chronic pain, similar patterns and comparable effect-sizes, as for the chronic pain analyses, were observed (Table 3).

## Physical exercise and number of pain locations

In general, women reported more pain locations than men irrespective of responses on the three exercise items. For exercise frequency, there was a clear dose-response association; the more frequent the exercise, the fewer the pain locations. This trend was particularly pronounced for female students (Fig 2—panels 2A and 2B).

Women reporting *moderate* exercise intensity had the least pain locations, compared to the women reporting little or hard exercise intensity. In contrast, exercise intensity was associated with pain location in a dose-response manner among males; the less exercise intensity, the more pain locations (Fig 2—panels 2C and 2D). In terms of exercise duration, students with

**Table 2. Association between physical exercise and chronic pain in male and female university and college and university students.**

| | n | (%) | ICD-11 chronic pain | | | | | |
|---|---|---|---|---|---|---|---|---|
| | | | Unadjusted model | | Adjusted model[§] | | Fully adjusted model[#] | |
| | n | (%) | OR | 95% CI | OR | 95% CI | OR | 95% CI |
| **Women** | | | | | | | | |
| **Physical exercise (frequency)** | | | | | | | | |
| Never/less than once a week | 2352 | (63.5) | 1.00 | | 1.00 | | 1.00 | |
| Once a week | 2501 | (61.7) | 0.92 | (0.83–1.01) | 0.94 | (0.85–1.04) | 0.96 | (0.90–1.00) |
| 2–3 times per week | 6719 | (59.6) | 0.85 | (0.78–0.92) | 0.88 | (0.81–0.96) | 0.92 | (0.84–0.99) |
| Almost every day | 3155 | (57.2) | 0.77 | (0.70–0.84) | 0.81 | (0.74–0.89) | 0.85 | (0.77–0.93) |
| **Physical exercise (intensity)** | | | | | | | | |
| I take it easy without breaking into a sweat or losing my breath | 2902 | (64.2) | 1.00 | | 1.00 | | 1.00 | |
| I push myself so hard that I lose my breath and break into a sweat | 10072 | (58.8) | 0.79 | (0.74–0.85) | 0.81 | (0.75–0.87) | 0.83 | (0.77–0.90) |
| I push myself to near-exhaustion | 1200 | (61.3) | 0.89 | (0.80–1.00) | 0.89 | (0.79–0.99) | 0.92 | (0.81–1.03) |
| **Physical exercise (duration)** | | | | | | | | |
| Less than 30 minutes | 1774 | (64.4) | 1.00 | | 1.00 | | 1.00 | |
| 30 minutes to 1 hour | 8034 | (59.6) | 0.80 | (0.72–0.88) | 0.81 | (0.74–0.89) | 0.83 | (0.76–0.92) |
| More than 1 hour | 4371 | (59.3) | 0.78 | (0.71–0.86) | 0.80 | (0.73–0.89) | 0.83 | (0.75–0.92) |
| **Men** | | | | | | | | |
| **Physical exercise (frequency)** | | | | | | | | |
| Never/less than once a week | 1151 | (49.5) | 1.00 | | 1.00 | | 1.00 | |
| Once a week | 758 | (44.0) | 0.79 | (0.69–0.90) | 0.81 | (0.70–0.92) | 0.83 | (0.73–0.95) |
| 2–3 times per week | 1955 | (41.3) | 0.70 | (0.63–0.78) | 0.71 | (0.64–0.79) | 0.74 | (0.66–0.82) |
| Almost every day | 1239 | (38.7) | 0.63 | (0.57–0.71) | 0.65 | (0.59–0.74) | 0.69 | (0.61–0.77) |
| **Physical exercise (intensity)** | | | | | | | | |
| I take it easy without breaking into a sweat or losing my breath | 764 | (46.6) | 1.00 | | 1.00 | | 1.00 | |
| I push myself so hard that I lose my breath and break into a sweat | 3232 | (41.8) | 0.82 | (0.73–0.92) | 0.84 | (0.75–0.94) | 0.85 | (0.76–0.86) |
| I push myself to near-exhaustion | 755 | (40.0) | 0.75 | (0.65–0.86) | 0.75 | (0.65–0.86) | 0.77 | (0.67–0.89) |
| **Physical exercise (duration)** | | | | | | | | |
| Less than 30 minutes | 540 | (45.3) | 1.00 | | 1.00 | | 1.00 | |
| 30 minutes to 1 hour | 2011 | (43.5) | 0.91 | (0.80–1.04) | 0.89 | (0.78–1.02) | 0.92 | (0.80–1.05) |
| More than 1 hour | 2200 | (40.5) | 0.80 | (0.70–0.91) | 0.80 | (0.70–0.92) | 0.83 | (0.72–0.95) |

[§] Adjusted for socio demographics, body-mass index, alcohol use and problems and sleep duration

[#] Additional adjustment for self-reported depression

short exercise durations (<30 minutes) had more pain locations than those exercising either 30 minutes to 60 minutes, or more than one hour a week (Fig 2—panels 2E and 2F).

## Discussion

This current study has several noteworthy findings. First, the prevalence of chronic pain was high, especially among women, with more than half of the students reporting at least one chronic pain location. With some gender differences, the overall pattern was an inverse dose-response association between exercise and chronic pain: the more frequent, harder or longer the exercise, the lower the odds of chronic pain. Similar findings were generally also observed for the number of pain locations: frequent exercise was associated with *fewer* pain locations. Adjusting for demographical, lifestyle factors and depression had little effect on the magnitude of the associations.

**Table 3. Association between physical exercise and moderate to severe chronic pain in male and female university and college and university students.**

| | ICD-11 moderate to severe chronic pain | | | | | | | |
| --- | --- | --- | --- | --- | --- | --- | --- | --- |
| | | | Unadjusted model | | Adjusted model[§] | | Fully adjusted model[#] | |
| | n | (%) | OR | 95% CI | OR | 95% CI | OR | 95% CI |
| **Women** | | | | | | | | |
| **Physical exercise (frequency)** | | | | | | | | |
| *Never/less than once a week* | 858 | (23.2) | 1.00 | | 1.00 | | 1.00 | |
| *Once a week* | 801 | (19.8) | 0.84 | (0.74–0.94) | 0.89 | (0.79–0.99) | 0.91 | (0.81–1.03) |
| *2–3 times per week* | 2124 | (18.8) | 0.79 | (0.71–0.87) | 0.85 | (0.77–0.94) | 0.90 | (0.81–0.99) |
| *Almost every day* | 918 | (16.6) | 0.67 | (0.60–0.75) | 0.74 | (0.67–0.83) | 0.79 | (0.70–0.88) |
| **Physical exercise (intensity)** | | | | | | | | |
| *I take it easy without breaking into a sweat or losing my breath* | 1037 | (23.0) | 1.00 | | 1.00 | | 1.00 | |
| *I push myself so hard that I lose my breath and break into a sweat* | 3043 | (17.8) | 0.74 | (0.68–0.80) | 0.76 | (0.69–0.82) | 0.79 | (0.72–0.86) |
| *I push myself to near-exhaustion* | 413 | (21.1) | 0.90 | (0.79–1.03) | 0.91 | (0.79–1.04) | 0.94 | (0.82–1.08) |
| **Physical exercise (duration)** | | | | | | | | |
| *Less than 30 minutes* | 634 | (23.0) | 1.00 | | 1.00 | | 1.00 | |
| *30 minutes to 1 hour* | 2419 | (18.0) | 0.74 | (0.66–0.82) | 0.77 | (0.69–0.86) | 0.80 | (0.72–0.89) |
| *More than 1 hour* | 1440 | (19.5) | 0.81 | (0.72–0.90) | 0.85 | (0.76–0.95) | 0.89 | (0.79–0.99) |
| **Men** | | | | | | | | |
| **Physical exercise (frequency)** | | | | | | | | |
| *Never/less than once a week* | 227 | (9.8) | 1.00 | | 1.00 | | 1.00 | |
| *Once a week* | 117 | (6.8) | 0.65 | (0.51–0.83) | 0.70 | (0.55–0.89) | 0.73 | (0.57–0.93) |
| *2–3 times per week* | 284 | (6.0) | 0.57 | (0.47–0.69) | 0.61 | (0.50–0.74) | 0.65 | (0.53–0.78) |
| *Almost every day* | 220 | (6.9) | 0.67 | (0.55–0.82) | 0.74 | (0.60–0.81) | 0.79 | (0.65–0.98) |
| **Physical exercise (intensity)** | | | | | | | | |
| *I take it easy without breaking into a sweat or losing my breath* | 141 | (8.6) | 1.00 | | 1.00 | | 1.00 | |
| *I push myself so hard that I lose my breath and break into a sweat* | 508 | (6.6) | 0.72 | (0.59–0.88) | 0.77 | (0.62–0.94) | 0.80 | (0.65–0.98) |
| *I push myself to near-exhaustion* | 133 | (7.0) | 0.76 | (0.59–0.98) | 0.79 | (0.61–1.03) | 0.83 | (0.64–1.08) |
| **Physical exercise (duration)** | | | | | | | | |
| *Less than 30 minutes* | 105 | (8.8) | 1.00 | | 1.00 | | 1.00 | |
| *30 minutes to 1 hour* | 319 | (6.9) | 0.75 | (0.59–0.95) | 0.75 | (0.59–0.86) | 0.79 | (0.62–1.00) |
| *More than 1 hour* | 360 | (6.6) | 0.70 | (0.56–0.89) | 0.72 | (0.56–0.91) | 0.76 | (0.60–0.97) |

[§] Adjusted for socio demographics, body-mass index, alcohol use and problems and sleep duration

[#] Additional adjustment for self-reported depression

The observed prevalence of chronic pain in the current study was even higher than what has been observed in similar studies. In a recent review and meta-analysis of chronic pain in epidemiological studies, a pooled chronic pain prevalence of 31% was reported, although the authors concluded that the lack of consistency in defining chronic pain makes evaluations and comparisons across study populations difficult [2]. A systematic review of previous studies reporting on the link between exercise and low back pain concluded that most studies in this field have failed to find a significant relationship between the two [40]. While this may indeed be the case for low back pain, it has been speculated that significant associations may be concealed due to crude measurements of both pain and exercise, as well as other methodological shortcomings [41]. To our knowledge, only Landmark and colleagues [9], using detailed exercise data from the large HUNT3 study from 2006–2008, have examined the link between frequency, duration, and intensity of recreational exercise and chronic pain. Although the HUNT3 study includes a substantial proportion of older adults (65+ years), subgroup analyses

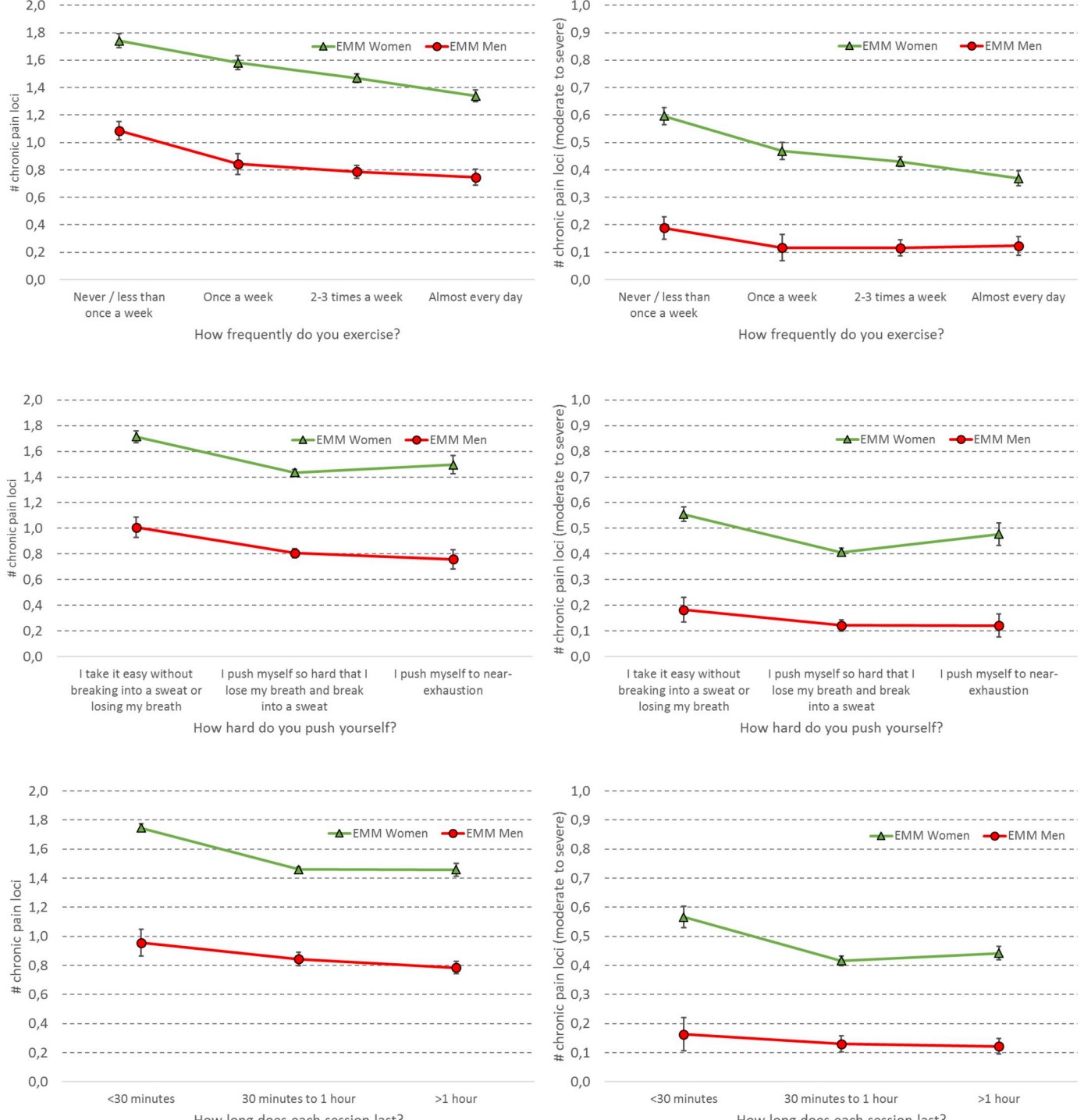

**Fig 2. Association between physical exercise frequency (top), intensity (middle) and duration (bottom) and number of ICD-11 chronic pain loci (left) and number of ICD-11 moderate to severe chronic pain loci (right) for male (red) and female (green) students at Norwegian colleges and universities.** Boxes represent estimated marginal means (EMM; adjusted for age), and error bars represent 95% confidence intervals.

of participants aged 20–64 years revealed a U-shaped association between exercise frequency and chronic pain. In contrast, the current study found this association to be linear; the more frequent the weekly exercise, the lower the risk of chronic pain. There may be several possibilities for these divergent findings, but we cannot disregard the possibility that differences in both sample composition and pain assessment, may play a role. Specifically, the assessment of chronic pain in the current study included a more thorough assessment, compared to the briefer 2-items pain inventory used in the HUNT3 study. Of importance, in a smaller longitudinal follow-up study of 6419 participants in the HUNT-3 study, Landmark et al. [10] found that regular exercise at baseline was associated with less pain over a 12 month follow-up period. However, the relationship was substantially reduced when controlling for baseline pain and was only significant for men. Concluding that the associations were close in time and weak, the Landmark et al. studies show that the significance of the exercise-pain link remains open for discussion. As such, the current national study of young adults, extending on previous evidence by using detailed instruments of both pain and exercise, suggest that there is indeed a significant association, and stronger in magnitude than previously believed, between reduced activity and risk of chronic pain.

The findings from the current study have some important clinical and public health implications. Both sedentary behavior [42] and pain [3] are some of our biggest public health challenges in the general population. The increasing level of inactivity has led the World Health Organization (WHO) to launch a global action plan [43] on physical activity for 2018 to 2030 in an attempt to make the world's population more active. This action plan aims at providing a system-based framework of effective and practical policy actions to countries in order to increase physical activity at all levels, emphasizing the need for a paradigm shift. Moreover, colleges and universities should to a larger extent, consider facilitating their students to take part in sports and exercise, perhaps also by having physical exercise become more integrated into the college environment.

In terms of future research, there is a need to conduct well-controlled and prospective studies to explore if, or to what extent, we can prevent chronic pain by increasing our level of physical exercise. It is also important to examine this across populations, both in terms of different age cohorts, and in healthy versus clinical samples. As also emphasized by Landmark et al. [10], identifying specific groups that may benefit more from exercise interventions to reduce the risk of chronic pain and improve health in general, would be an important objective for future investigations.

## Methodological considerations

The most important limitation of the current study is the cross-sectional nature of the study, limiting our ability to study the directionality between physical exercise and chronic pain. As such, inactivity may be both a risk factor, as well as a consequence of chronic pain. Another important limitation is the modest response rate, with little information about the characteristics of non-participants beyond age and gender distribution. Selective participation could bias the prevalence observed to the extent the selection was correlated with reports of chronic pain. On the one hand, it has been shown that non-participants of health surveys in general have poorer health than participants [44]. The current results may, therefore, represent an underestimation of the true prevalence of chronic pain in the target population. On the other hand, people are in general more prone to participate in a survey if the topic is relevant to them personally [45]. As the information material of the SHoT2018-study focused much on "how the students *really* are and feel", one may speculate if this would lead to a higher participation rate of individuals who felt that the topic was of particular relevance to them. Since response rates

are particularly important in prevalence studies, care should be taken when generalizing the current findings to the whole student population. Rather, it may be more appropriate to emphasize the relative differences between men and women, as these estimates are less prone to selection bias. Using a web-based survey approach may have contributed to the modest response rate, as electronic platforms have been shown to yield somewhat lower participation rates compared to traditional approaches [46, 47]. However, there are also reports showing similar response rates between online and paper questionnaires [48]. A final limitation related to the self-reported physical activity measure is that it is more accurate to say that we assessed *perceived* intensity, as less fit individuals will feel exhausted by an intensity that a fit person will feel comfortable.

Strengths of the current study include the large and heterogeneous sample, the use of well-validated instruments, and the inclusion of several potential confounders.

## Conclusions

The demonstrated health benefits of regular exercise suggest that facilitating young adults to become more physically active should be a prioritized task both for political and educational institutions.

## Acknowledgments

We wish to thank all students participating in the study, as well as the three largest student welfare organizations in Norway (SiO, Sammen, and SiT), who initiated and designed SHoT study.

## Author Contributions

**Conceptualization:** Børge Sivertsen.

**Data curation:** Kari Jussie Lønning, Børge Sivertsen.

**Formal analysis:** Michael Grasdalsmoen, Børge Sivertsen.

**Funding acquisition:** Kari Jussie Lønning, Børge Sivertsen.

**Methodology:** Michael Grasdalsmoen, Bo Engdahl, Mats K. Fjeld, Ólöf A. Steingrímsdóttir, Christopher S. Nielsen, Hege R. Eriksen, Børge Sivertsen.

**Project administration:** Kari Jussie Lønning.

**Supervision:** Børge Sivertsen.

**Writing – original draft:** Michael Grasdalsmoen, Børge Sivertsen.

**Writing – review & editing:** Bo Engdahl, Mats K. Fjeld, Ólöf A. Steingrímsdóttir, Christopher S. Nielsen, Hege R. Eriksen, Kari Jussie Lønning.

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
