## [Decision Letter · Decision Letter 0]

4 May 2020

PONE-D-20-09746

Physical exercise and chronic pain in university students

PLOS ONE

Dear Prof. Sivertsen,

Thank you for submitting your manuscript to PLOS ONE. After careful consideration, we feel that it has merit but does not fully meet PLOS ONE’s publication criteria as it currently stands. Therefore, we invite you to submit a revised version of the manuscript that addresses the points raised during the review process.

I agreed with the reviewers that there is need to strengthen the manuscript by attending to some methodological analytical omissions. Specifically, there is need to include information on the utility and psychometrics of the measurement scales in the studied population, improve clarity on the physical activity continuum (physical inactivity vs sedentary time) and the language of the survey and its administration, It is also very important to reconsider the analytic method or provide strong justification for the current analyses. Why was socioeconomic status and psychological variables (e.g., depression measures or other mental health scales) not adjusted for in the analyses considering these are potential mediators of chronic pain in the Scandinavian?   

We would appreciate receiving your revised manuscript by Jun 18 2020 11:59PM. To enhance the reproducibility of your results, we recommend that if applicable you deposit your laboratory protocols in protocols.io, where a protocol can be assigned its own identifier (DOI) such that it can be cited independently in the future. For instructions see: http://journals.plos.org/plosone/s/submission-guidelines#loc-laboratory-protocols

We look forward to receiving your revised manuscript.

Kind regards,

Adewale L. Oyeyemi, Ph.D

Academic Editor

PLOS ONE

2. Please modify the title to ensure that it is meeting PLOS’ guidelines (https://journals.plos.org/plosone/s/submission-guidelines#loc-title). In particular, the title should be "specific, descriptive, concise, and comprehensible to readers outside the field" and in this case it is not informative and specific about your study's scope and methodology.

Reviewers' comments:

Reviewer's Responses to Questions

**Comments to the Author**

1. Is the manuscript technically sound, and do the data support the conclusions?

Reviewer #1: Yes

Reviewer #2: Yes

2. Has the statistical analysis been performed appropriately and rigorously? 

Reviewer #1: Yes

Reviewer #2: I Don't Know

3. Have the authors made all data underlying the findings in their manuscript fully available?

Reviewer #1: Yes

Reviewer #2: Yes

4. Is the manuscript presented in an intelligible fashion and written in standard English?

Reviewer #1: Yes

Reviewer #2: Yes

5. Review Comments to the Author

Reviewer #1: Re: comments on the manuscript number “PONE-D-20-09746”.

Importance and objective of the paper

Chronic pain is a major health problem globally. It is highlighted as one of the leading cause of years lived with disability, imposing substantial burdens on individuals and community. Although there are small percent of people develop chronic pain, the treatment for those people are complicated and costly. The beneficial effects of physical activity for many chronic diseases have been well documented. Therefore, I see this article is important as it investigates the association between engaging in different amounts of physical exercise and chronic pain.

Generally, the authors do a good job in describing the problem and research question, detailing the methods and results, and providing contextual information to aid in the interpretation of the results. However, there is room for improvement.

Major comment:

I see that the authors of this study are frequently using passive voice in writing the paper. I think that the passive voice is weak and sometime incorrect. I would suggest using active voice in some sections of the paper.

Minor comments:

INTRODUCTION

1) “In terms of the latter, there is some evidence suggesting a link between a sedentary lifestyle and the development of chronic pain”

I would suggest talking about lower levels of physical activity or inactivity, which is related to your topic, instead of talking about sedentary behavior. “Sedentary behavior is any waking behavior characterized by an energy expenditure ≤1.5 metabolic equivalents (METs), while in a sitting, reclining or lying posture”. Therefore, it is likely for someone to accumulate large amounts of both vigorous physical activity and sedentary behavior in one day. For example, someone may work 8 hours on desk and in the same day he may run or swim for 2 hours.

I would recommend you reading these articles: https://ijbnpa.biomedcentral.com/articles/10.1186/s12966-017-0525-8

https://www.nrcresearchpress.com/doi/10.1139/h2012-024#.XpWFA8gzaUk

I would suggest for you to rewrite this paragraph and focus on studies that measured the association between physical activity levels and pain.

METHODS

1) Procedure

The procedure is confusing me! As I see that you are talking about a different study published in 2018! I was expecting to find some information on the procedures that you followed in your study. I think you need some extra sentences in the beginning of this section before starting to talk about SHoT2018.

2) Physical exercise

“In the current study, the response options “Never” and “Less than once a week” were combined for the frequency item, constituting the reference category.”.

As I see in the results section that the highest physical activity category was set to be the reference category. Therefore, I see conflicted information in this study! Could you clarify that?

I would suggest that you set the lowest physical activity category to be the reference category. It is very important for the readers and researchers to look at whether increasing physical activity is associated with any additional benefits for chronic pain.

3) Sociodemographic information

“economically active” (annual income > 10,000 NOK) versus “economically inactive” (< 10,000 NOK).”

So, what about people who will have income equal to 10,000! They will be categorized as economically active or inactive?

4) Sleep duration

What do you mean with rise time? Do you mean wake time?

RESULTS

1) In general, I suggest for you to set up the lowest category to be the reference category.

2) “Table 1. Sociodemographic and clinical characteristics of the SHoT 2018 study.”

I would suggest for you talking about the data that you used in your study. You could only clarify in the procedure section that you used data from the SHoT2018. Could you do that throughout the manuscript?

3) Physical exercise and chronic pain

I see that in this section that you are sometimes talking about chronic pain intensity and sometimes chronic pain locations! However, when I looked at the table I found “chronic pain” only! Could you be clearer in your writing about this point?

4) Physical exercise and moderate to severe chronic pain

“Also, the ORs for exercise duration and moderate-to-severe chronic pain among males were higher than those found for chronic pain.”

I did not see big differences between them! I would suggest for you to focus on the significant results.

DISCUSSION

1) “This large national health survey from 2018, inviting all Norwegian full-time college and university students in the age 18-35, has several interesting findings.”

It does not make sense to me whether you are talking about your study findings or the survey!

2) “The observed prevalence of chronic pain in the current study was high but comparable to what

has also been observed in similar studies. In a recent review and meta-analysis of chronic pain in

epidemiological studies, a pooled chronic pain prevalence of 31% was reported, although the authors concluded that the lack of consistency in defining chronic pain makes evaluations and comparisons across study populations difficult”.

The prevalence of chronic pain in your study was around 54%! I do not think it is similar to the review that your reported which says 31%!

3) “Strengths of the SHoT study include the very large sample size, in combination with several well-validated questionnaires.”.

I think this sentence is related to the next paragraph!

4) Could you write a conclusion paragraph at the end of the DISCUSSION section?

Reviewer #2: Although this paper is very well written and addresses an important topic, my biggest concern is whether psychological factors and economic factors have been considered as confounders among this group. This group is particularly at risk of depression due to financial issues (not sure if same in Norway as other countries) or family issues, and there is a known link between depression and chronic pain, and if not factored in could have influenced the results. In addition, Scandinavian countries are known for having a high prevalence of depression - due to factors such as weather, etc. Has this been factored in as confounders ? I see you have included economic status, but was this adjusted for as well? No psychological outcomes seem to have been measured.

And then one other issue - in which language where all questionnaires/scales administered? Were those not previously validated and translated, translated and validated at any point among this group in this study? not clear from text, although reference made for certain tools.

6. PLOS authors have the option to publish the peer review history of their article (what does this mean?). If published, this will include your full peer review and any attached files.

Reviewer #1: Yes: Hosam Alzahrani

Reviewer #2: No

---

## [Author Response · Author response to Decision Letter 0]

20 May 2020

Dear Editor

Thank you for the constructive and positive comments from you and the two reviewers, and our chance to revise and improve our manuscript. As outlined below, we have responded to each of the comments and we have described the changes we have made in the manuscript. We hope you agree that the manuscript has improved through this process, and we are pleased to submit our revised manuscript for your consideration. 

EDITOR

I agreed with the reviewers that there is need to strengthen the manuscript by attending to some methodological analytical omissions. Specifically, there is need to include information on the utility and psychometrics of the measurement scales in the studied population, improve clarity on the physical activity continuum (physical inactivity vs sedentary time) and the language of the survey and its administration, It is also very important to reconsider the analytic method or provide strong justification for the current analyses. Why was socioeconomic status and psychological variables (e.g., depression measures or other mental health scales) not adjusted for in the analyses considering these are potential mediators of chronic pain in the Scandinavian? 

Response: We agree with these comments, and as outlined in our responses to the reviewers below, we have modified the manuscript to accommodate these issues. Specifically, we now provide more details regarding the psychometrics of the physical exercise questionnaire, and we no longer refer to the term “sedentary behaviors” (rather, we use physical activity or exercise throughout the manuscript. We now also state that all questionnaires in the SHOT2018 were administered in Norwegian, and questionnaires not previously translated into Norwegian were translated, and then back-translated to the original language (English), to ensure accuracy. As also suggested by reviewer #2, we now also control for depression (and income) in the fully adjusted logistic regression analyses. 

REVIEWER # 1

Importance and objective of the paper

Chronic pain is a major health problem globally. It is highlighted as one of the leading cause of years lived with disability, imposing substantial burdens on individuals and community. Although there are small percent of people develop chronic pain, the treatment for those people are complicated and costly. The beneficial effects of physical activity for many chronic diseases have been well documented. Therefore, I see this article is important as it investigates the association between engaging in different amounts of physical exercise and chronic pain.

Generally, the authors do a good job in describing the problem and research question, detailing the methods and results, and providing contextual information to aid in the interpretation of the results. However, there is room for improvement.

Major comment:

I see that the authors of this study are frequently using passive voice in writing the paper. I think that the passive voice is weak and sometime incorrect. I would suggest using active voice in some sections of the paper.

Response: We now use more active voice in the manuscript.

Minor comments:

INTRODUCTION

1) “In terms of the latter, there is some evidence suggesting a link between a sedentary lifestyle and the development of chronic pain”

I would suggest talking about lower levels of physical activity or inactivity, which is related to your topic, instead of talking about sedentary behavior. “Sedentary behavior is any waking behavior characterized by an energy expenditure ≤1.5 metabolic equivalents (METs), while in a sitting, reclining or lying posture”. Therefore, it is likely for someone to accumulate large amounts of both vigorous physical activity and sedentary behavior in one day. For example, someone may work 8 hours on desk and in the same day he may run or swim for 2 hours.

I would recommend you reading these articles: 

https://ijbnpa.biomedcentral.com/articles/10.1186/s12966-017-0525-8

https://www.nrcresearchpress.com/doi/10.1139/h2012-024#.XpWFA8gzaUk

I would suggest for you to rewrite this paragraph and focus on studies that measured the association between physical activity levels and pain.

Response: We appreciate the reviewer pointing us to these two interesting papers. We have now rewritten the relevant paragraph in the introduction accordingly, no longer using the term sedentary/sedentary behaviors. 

METHODS

1) Procedure

The procedure is confusing me! As I see that you are talking about a different study published in 2018! I was expecting to find some information on the procedures that you followed in your study. I think you need some extra sentences in the beginning of this section before starting to talk about SHoT2018.

Response: We agree that this section was not entirely clear with regards to the description of the entire SHoT018 study and the current paper. We have now modified this section to improve clarity. 

2) Physical exercise

“In the current study, the response options “Never” and “Less than once a week” were combined for the frequency item, constituting the reference category.”.

As I see in the results section that the highest physical activity category was set to be the reference category. Therefore, I see conflicted information in this study! Could you clarify that?

I would suggest that you set the lowest physical activity category to be the reference category. It is very important for the readers and researchers to look at whether increasing physical activity is associated with any additional benefits for chronic pain.

Response: We appreciate these comments and for pointing out the inconsistencies regarding what is the reference group in the physical exercise measure. As suggested by the reviewer, we now use the lowest level of physical activity as the reference category in order to more clearly show the beneficial/protective effect (shown by OR < 1.0) of increasing exercise on chronic pain. All relevant text and tables have been modified accordingly. 

3) Sociodemographic information

“economically active” (annual income > 10,000 NOK) versus “economically inactive” (< 10,000 NOK).”

So, what about people who will have income equal to 10,000! They will be categorized as economically active or inactive?

Response: We have now fixed this error to “economically inactive” (≤ 10,000 NOK).”

4) Sleep duration

What do you mean with rise time? Do you mean wake time?

Response: We now use the term wake up time instead of rise time. 

RESULTS

1) In general, I suggest for you to set up the lowest category to be the reference category.

Response: We agree, and this has now been changed throughout the paper. 

2) “Table 1. Sociodemographic and clinical characteristics of the SHoT 2018 study.”

I would suggest for you talking about the data that you used in your study. You could only clarify in the procedure section that you used data from the SHoT2018. Could you do that throughout the manuscript?

Response: This has now been changed throughout the manuscript.

3) Physical exercise and chronic pain

I see that in this section that you are sometimes talking about chronic pain intensity and sometimes chronic pain locations! However, when I looked at the table I found “chronic pain” only! Could you be clearer in your writing about this point?

Response: We agree that this terminology here may be somewhat confusing. The term “Chronic pain” (2nd row in Table 2) and ”Moderate to severe chronic pain” ” (2nd row in Table 3) refers to the two ICD-11 definitions of outlined in the instrument section in the Methods. We have now inserted “ICD-11” to these two rows to improve clarity. 

4) Physical exercise and moderate to severe chronic pain

“Also, the ORs for exercise duration and moderate-to-severe chronic pain among males were higher than those found for chronic pain.”

I did not see big differences between them! I would suggest for you to focus on the significant results.

Response: True. This sentence has now been deleted. 

DISCUSSION

1) “This large national health survey from 2018, inviting all Norwegian full-time college and university students in the age 18-35, has several interesting findings.”

It does not make sense to me whether you are talking about your study findings or the survey!

Response: We have now removed this sentence from the beginning of the Discussion.

2) “The observed prevalence of chronic pain in the current study was high but comparable to what has also been observed in similar studies. In a recent review and meta-analysis of chronic pain in epidemiological studies, a pooled chronic pain prevalence of 31% was reported, although the authors concluded that the lack of consistency in defining chronic pain makes evaluations and comparisons across study populations difficult”.

The prevalence of chronic pain in your study was around 54%! I do not think it is similar to the review that your reported which says 31%!

Response: We agree, and we have now changed this sentence to: 

“The observed prevalence of chronic pain in the current study was even higher than what has been observed in similar studies.”

3) “Strengths of the SHoT study include the very large sample size, in combination with several well-validated questionnaires.”.

I think this sentence is related to the next paragraph!

Response: True. The paragraph outlining the study strengths have been modified accordingly. 

4) Could you write a conclusion paragraph at the end of the DISCUSSION section?

Response: We have now added the following conclusion at the end of the discussion: 

“The demonstrated health benefits of regular exercise suggest that facilitating young adults to become more physically active should be a prioritized task both for political and educational institutions.”

REVIEWER # 2

Although this paper is very well written and addresses an important topic, my biggest concern is whether psychological factors and economic factors have been considered as confounders among this group. This group is particularly at risk of depression due to financial issues (not sure if same in Norway as other countries) or family issues, and there is a known link between depression and chronic pain, and if not factored in could have influenced the results. In addition, Scandinavian countries are known for having a high prevalence of depression - due to factors such as weather, etc. Has this been factored in as confounders ? I see you have included economic status, but was this adjusted for as well? No psychological outcomes seem to have been measured.

Response: This is an important point, and we now include self-reported depression in the list of confounders. We have updated all relevant parts of the manuscript to reflect this change. Specifically related to the Tables, we have added a third column/Model to Table 2 and 3, labeled “Fully adjusted Model” in which we now additionally adjust for depression, plus the other confounders listed under “Adjusted Model”. 

And then one other issue - in which language where all questionnaires/scales administered? Were those not previously validated and translated, translated and validated at any point among this group in this study? not clear from text, although reference made for certain tools.

Response: Where possible, questionnaires were administered using the validated Norwegian translation (e.g. physical exercise, sleep duration and AUDIT. Other questionnaires not previously translated into Norwegian were translated, and then back-translated to the original language (English), to ensure accuracy. This information has now been added to the Methods section. For the GRIP instrument, we also added the following text:

“Instructions and questions in GRIP were put in Norwegian. Translation to English was made by a certified translator, but back translation is still in process [16]”.

---

## [Decision Letter · Decision Letter 1]

16 Jun 2020

Physical exercise and chronic pain in university students

PONE-D-20-09746R1

Dear Dr. Sivertsen,

We’re pleased to inform you that your manuscript has been judged scientifically suitable for publication and will be formally accepted for publication once it meets all outstanding technical requirements.

Kind regards,

Adewale L. Oyeyemi, Ph.D

Academic Editor

PLOS ONE

Additional Editor Comments (optional):

Reviewers' comments:

Reviewer's Responses to Questions

**Comments to the Author**

1. If the authors have adequately addressed your comments raised in a previous round of review and you feel that this manuscript is now acceptable for publication, you may indicate that here to bypass the “Comments to the Author” section, enter your conflict of interest statement in the “Confidential to Editor” section, and submit your "Accept" recommendation.

Reviewer #1: All comments have been addressed

Reviewer #2: All comments have been addressed

2. Is the manuscript technically sound, and do the data support the conclusions?

Reviewer #1: Yes

Reviewer #2: Yes

3. Has the statistical analysis been performed appropriately and rigorously? 

Reviewer #1: Yes

Reviewer #2: Yes

4. Have the authors made all data underlying the findings in their manuscript fully available?

Reviewer #1: Yes

Reviewer #2: Yes

5. Is the manuscript presented in an intelligible fashion and written in standard English?

Reviewer #1: Yes

Reviewer #2: Yes

6. Review Comments to the Author

Reviewer #1: (No Response)

Reviewer #2: (No Response)

7. PLOS authors have the option to publish the peer review history of their article (what does this mean?). If published, this will include your full peer review and any attached files.

Reviewer #1: No

Reviewer #2: No

---

## [Editor Report · Acceptance letter]

18 Jun 2020

PONE-D-20-09746R1 

Physical exercise and chronic pain in university students 

Dear Dr. Sivertsen:

I'm pleased to inform you that your manuscript has been deemed suitable for publication in PLOS ONE. Congratulations! Your manuscript is now with our production department. 

Kind regards, 

on behalf of

Dr. Adewale L. Oyeyemi 

Academic Editor

PLOS ONE